# Local large temperature difference and ultra-wideband photothermoelectric response of the silver nanostructure film/carbon nanotube film heterostructure

Bocheng Lv[1], Yu Liu[2], Weidong Wu[3], Yan Xie[3], Jia-Lin Zhu[1], Yang Cao[4], Wanyun Ma[1], Ning Yang[5], Weidong Chu[5], Yi Jia[6], Jinquan Wei [7✉] & Jia-Lin Sun [1✉]

Photothermoelectric materials have important applications in many fields. Here, we joined a silver nanostructure film and a carbon nanotube film by van der Waals force to form a heterojunction, which shows excellent photothermal and photoelectric conversion properties. The local temperature difference and the output photovoltage increase rapidly when the heterojunction is irradiated by lasers with wavelengths ranging from ultraviolet to terahertz. The maximum temperature difference reaches 215.9 K, which is significantly higher than that of other photothermoelectric materials reported in the literature. The photothermal and photoelectric responsivity depend on the wavelength of lasers, which are 175~601 K W$^{-1}$ and 9.35~40.4 mV W$^{-1}$, respectively. We demonstrate that light absorption of the carbon nanotube is enhanced by local surface plasmons, and the output photovoltage is dominated by Seebeck effect. The proposed heterostructure can be used as high-efficiency sensitive photothermal materials or as ultra-wideband fast-response photoelectric materials.

[1] State Key Laboratory of Low-Dimensional Quantum Physics, Department of Physics, Tsinghua University, 100084 Beijing, China. [2] College of Mechanical Engineering and Automation, Fuzhou University, 350108 Fuzhou, China. [3] Department of Engineering Physics, Tsinghua University, 100084 Beijing, China. [4] School of Instrumentation Science and Opto-electronics Engineering, Beijing Information Science & Technology University, 100192 Beijing, China. [5] Institute of Applied Physics and Computational Mathematics, 100088 Beijing, China. [6] Qian Xuesen Laboratory of Space Technology, China Academy of Space Technology, 100094 Beijing, China. [7] Key Lab for Advanced Materials Processing Technology of Education Ministry, School of Materials Science and Engineering, Tsinghua University, 100084 Beijing, China. ✉email: jqwei@tsinghua.edu.cn; jlsun@tsinghua.edu.cn

In recent years, with the rapid development of economy, the greenhouse gases emitted by burning fossil fuels have led to a rise of the global temperature[1]. Therefore, it is urgently required for developing clean and renewable energy conversion technologies[2]. Much attention has been attracted to using the photothermoelectric effect to realize light–heat–electricity energy conversion. The photothermoelectric effect refers to the temperature difference ($\Delta_T$) formed at different locations of the device after absorption of photon energy, which is then converted into an output voltage through the Seebeck effect[3]. In the photothermoelectric effect, the output voltage is related to the $\Delta_T$ and the Seebeck coefficient of the material. The magnitude of the $\Delta_T$ is determined by the light absorption and heat capacity of materials, while the Seebeck coefficient is determined by the density of states near the Fermi energy level[4]. Some photothermoelectric materials and devices with good performance have been prepared. For example, when a laser with a wavelength of 906 nm and a power of 70 mW was used to irradiate the interface between monolayer graphene and a gold electrode using a Cr/SiO$_2$/Cr/polyethylene terephthalate (PET) as the substrate, the temperature of the device increased by 37 K[5]. However, when the substrate was directly irradiated by the same laser, the temperature of the Cr/SiO$_2$/Cr/PET increased from 26.7 to 190.7 °C (i.e., $\Delta T = 164$ K)[5]. When a three-dimensional porous graphene was irradiated by a terahertz laser (wavelength of 118.8 μm) with a power density of 19 mW mm$^{-2}$, its temperature increased by 109.3 K[6]. In addition, some photothermal materials show promise in solar energy harvesting[7], making microcontrollers[8] and cancer treatment[9]. For example, Ti$_2$O$_3$ nanoparticles can be used as excellent materials for solar energy collection and seawater desalination. When exposed to 1 kW m$^{-2}$ simulated sunlight, the temperature of Ti$_2$O$_3$ increases by 24.5 K after about 200 s[10]. Single-walled carbon nanotube (CNT) with poly(3-hexylthiophene) dispersed in PDMS sheets (P3HT-SWNT-PDMS) are also excellent photothermal conversion materials, and their temperature can be increased by 45 K after 180 s when irradiated by a laser with a wavelength of 785 nm at 80 mW mm$^{-2}$ [11].

It has been shown that photons can excite the strong local field on the surface of noble-metal nanomaterials, which can effectively enhance absorption of photons and improve the light–heat–electricity energy conversion efficiency of photothermoelectric materials[12–17]. We also found that it can produce strong local field on silver nanostructure when they are irradiated by a laser[12,13], which is beneficial to the enhancement of Raman scattering effect[15,16] or negative photoconductance effect[12,13].

CNTs have an ultra-wideband absorption spectrum[18–25], high charge-carrier mobility[26], and good light absorption[27]. Previously, we used a suspended CNT film (CNTF) as a photodetector[28], which showed ultra-wideband response from ultraviolet to terahertz wavelengths, and a high light responsivity of 25.4 mA W$^{-1}$ at 1064 nm and a power of ~24 mW. Based on the above factors, we joined an Ag nanostructure film (AgNSF) and a CNTF to fabricate an AgNSF/CNTF heterostructure on a glass substrate by van der Waals force, and then investigated its photothermoelectric conversion and response performance. The sample showed amazing photothermal conversion ability and good photoelectric response characteristics. When a laser irradiates the AgNSF/CNTF heterojunction, the temperature at the center of the heterojunction and the output photovoltage of the device increased rapidly in the ultra-wideband range from ultraviolet to terahertz wavelengths, and the photothermal and photoelectric response time was on an order of tens of milliseconds. The experimental data showed that the AgNSF/CNTF heterostructure can be used as a material for making fast-response ultra-wideband photodetectors, and it also can be used as a material for producing high-efficiency sensitive solar collectors or photothermal materials.

## Results

**Local large $\Delta_T$ on the AgNSF/CNTF heterojunction sample.** Heatmap videos of the sample are recorded by an infrared camera (see Supplementary Movies 1–6) during laser irradiation in real time. The temperature of the heterojunction increase clearly when it is irradiated by lasers with different wavelengths. For example, a representative heatmap is shown in Fig. 1a. When the AgNSF/CNTF heterojunction was irradiated by a laser with a wavelength of 1064 nm and a power of 518 mW, within the first few tens of milliseconds of laser irradiation, the local temperature increases sharply to 512.15 K, forming a large $\Delta_T$ of 215.9 K, which is intuitively shown by a three-dimensional heatmap in the inset of Fig. 1a. The $\Delta_T$ of 215.9 K is significantly higher than the maximum $\Delta_T$ achieved by photothermoelectric (or photothermal) conversion materials reported in literatures (see Supplementary Table 1). The melting point of silver nanowires is 848 K[29], and the CNTF was tested under atmospheric conditions, which can maintain unchanged at high temperature of 873 K. Therefore, if we continued to increase the power of the laser irradiating the sample, the temperature at the heterojunction would have a large space to increase. The corresponding relationship between the

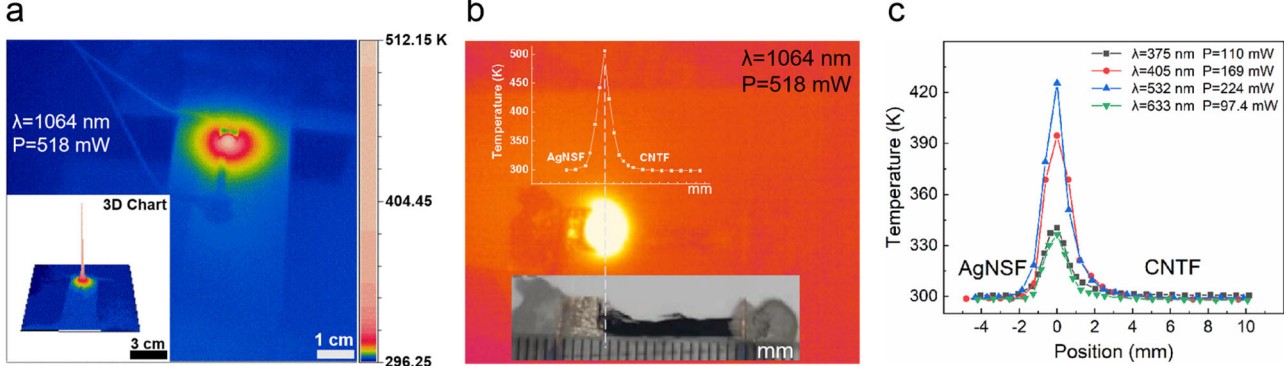

**Fig. 1 Local large $\Delta_T$. a** Heatmap obtained when the heterojunction of the sample was irradiated by a laser with wavelength of 1064 nm and power of 518 mW. The inset shows a three-dimensional image of the heat distribution. **b** The infrared photograph of the actual sample. Upper inset is temperature–position curve, in which the location with the highest temperature is the center of the heterojunction; lower inset is a photograph of the actual sample. **c** Temperature–position curves of the AgNSF/CNTF heterojunction irradiated by lasers with wavelengths of 375, 405, 532, and 633 nm, where the position of 0 mm is the center of the heterojunction.

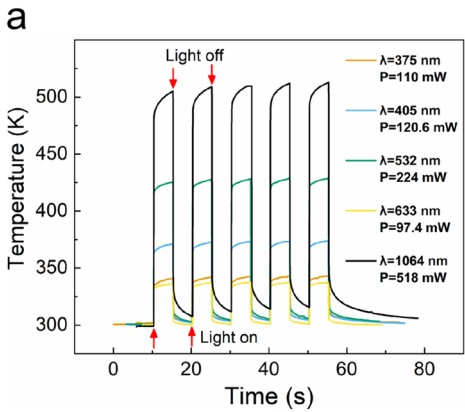
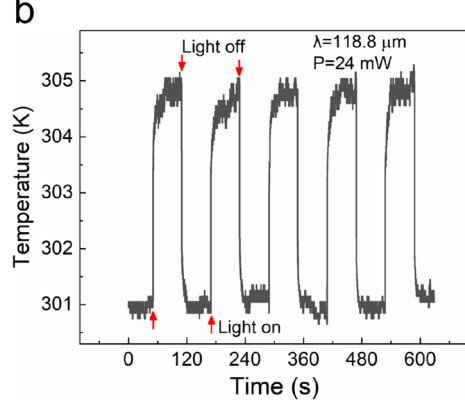

**Fig. 2 Temperature–time curves.** Photothermal response curves of the center temperature of the heterojunction with time when the heterojunction was irradiated by lasers with wavelengths of **a** 375 nm, 405 nm, 532 nm, 633 nm, 1064 nm, and **b** 118.8 μm, respectively.

photograph (lower inset) and the infrared photograph of the actual sample is shown in Fig. 1b. When the center temperature of the heterojunction reaches 512.15 K, we simultaneously record the temperature of both the AgNSF and CNTF sides by an infrared camera. The upper inset in Fig. 1b shows the temperature–position curve. Similar temperature–position curves were obtained when lasers with other wavelengths were used to irradiate the heterojunction, as shown in Fig. 1c.

**Rapid photothermal response of the AgNSF/CNTF heterojunction.** To further study the response speed and stability of the photothermal conversion of the AgNSF/CNTF heterojunction, we derived the temperature–time curves at the heterojunction with and without laser irradiation from the thermal distribution videos (Supplementary Movie 1, 375 nm, 110 mW; Supplementary Movie 2, 405 nm, 120.6 mW; Supplementary Movie 3, 532 nm, 224 mW; Supplementary Movie 4, 633 nm, 97.4 mW; Supplementary Movie 5, 1064 nm, 518 mW; Supplementary Movie 6, 118.8 μm, 24 mW) recorded with an infrared camera.

From the temperature–time curves in Fig. 2, the center temperature of the AgNSF/CNTF heterojunction showed an obvious rapid response process when it was irradiated by lasers with different wavelengths (from ultraviolet to terahertz).

Here, we define a photothermal responsivity $R_T$ as the ratio of the final temperature increase at the center of the heterojunction to the absorbed light power after laser irradiation on the heterojunction:

$$R_T = \frac{|\Delta T|}{P_{in} - P_t} \quad (1)$$

where $\Delta T$ is the change in temperature, $P_{in}$ is the incident optical power (measured by a power meter in front of the sample), and $P_t$ is the transmitted optical power (measured by a power meter behind the sample).

For the response time, the rise time refers to the time taken to increase from the minimum value to 80% of the maximum value ($\Delta T$), while the fall time is the time taken for $\Delta T$ to decrease from the maximum value to 20% of the maximum value[30].

The performance parameters of photothermal conversion of the AgNSF/CNTF heterojunction irradiated by lasers with different wavelengths are given in supplementary Table 2. The photothermal responsivity was in the range 175–601 K W$^{-1}$, and the response time was tens of milliseconds except for irradiation by a laser with wavelength of 118.8 μm.

**Ultra-wideband photoelectric response of the AgNSF/CNTF heterojunction.** Figure 3a is the setup used to measure

current–voltage (I–V) curves of the sample. Figure 3b shows the I–V curve when the heterojunction is irradiated by a laser with wavelength of 1064 nm. The I–V curves of the heterojunction irradiated by lasers with other wavelengths showed similar trends. Figure 3c shows the photocurrent-voltage (ΔI–V) curves in the same coordinate system for the heterojunction irradiated by lasers with wavelengths ranging from 375 nm to 118.8 μm. The photocurrent is defined as $\Delta I = I_{light} - I_{dark}$, where $I_{light}$ is the current measured with laser illumination and $I_{dark}$ is the current measured at dark without laser illumination.

The I–V curves of the sample were linear (Fig. 3b), which indicates that the AgNSF and CNTF is in ohmic contact with a resistance of ~30 Ω. In the first quadrant, that is, when the bias voltage is positive, the current of the sample irradiated by the laser was higher than the case without illumination. In the third quadrant, that is, when the bias voltage was negative, the absolute value of current decreases after illuminating the heterojunction. The direction of the photovoltage generated inside the sample is from the AgNSF side to the CNTF side after the laser illumination. In the I–V curves, the absolute value of the intersection point of the red curve and the horizontal axis is the photovoltage generated by laser irradiation on the heterojunction. The vertical coordinate of the intersection point of the red curve and the vertical axis is the photocurrent generated by laser irradiation of the heterojunction. From Fig. 3c, photocurrent ΔI generates in the sample when it is irradiated by lasers with various wavelengths. In addition, the photocurrent ΔI almost independent on the applied bias by fixing the wavelength and power of laser illumination.

The photovoltage–time response curves measured by the oscilloscope are shown in Fig. 3d-e. We calculate the photoelectric responsivity ($R_V$) of the sample by

$$R_V = \frac{|\Delta U|}{P_{in} - P_t} \quad (2)$$

where $P_{in}$ is the incident optical power, $P_t$ is the transmitted optical power, and $\Delta U$ is the output photovoltage value of the sample.

For the response time, the rise time refers to the time taken for the photovoltage to increase from the minimum to 80% of the maximum value, while the fall time refers to the time taken for the photovoltage to decrease from the maximum value to 20% of the maximum value[30].

The equivalent noise power (NEP), which is also known as the minimum metering power, can be expressed as

$$NEP = \frac{\sqrt{4k_B TR}}{R_V} \quad (3)$$

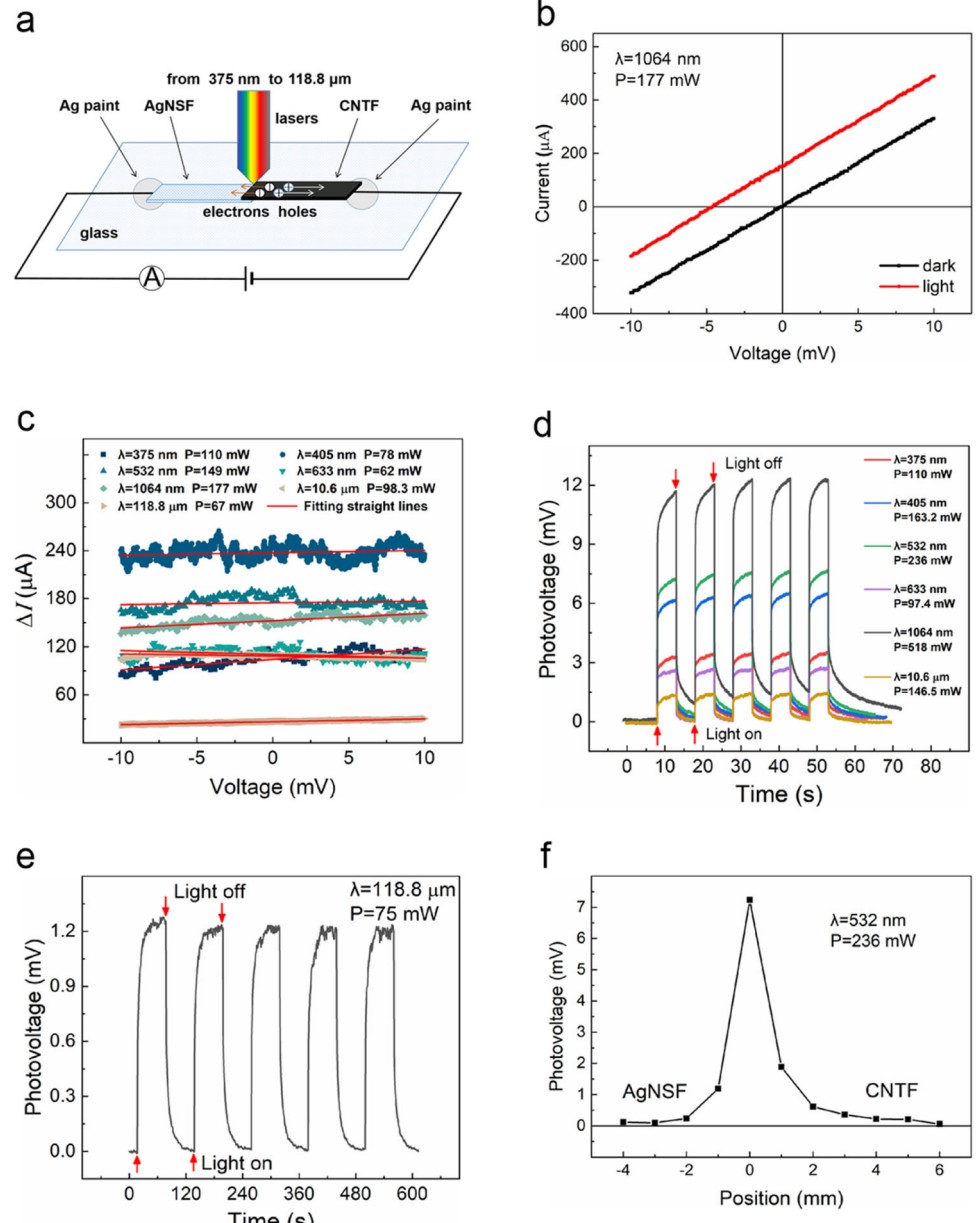

**Fig. 3 Photoelectric performance. a** Schematic diagram of the device used to measure the *I–V* curves of the sample. **b** *I–V* curve when the heterojunction irradiated by a laser with wavelength of 1064 nm. **c** Δ*I–V* curves of the heterojunction irradiated by lasers with wavelengths ranging from 375 nm to 118.8 μm. Photovoltage–time response curves of the AgNSF/CNTF sample irradiated by lasers with wavelengths of **d** 375 nm, 405 nm, 532 nm, 633 nm, 1064 nm, 10.6 μm and **e** 118.8 μm, respectively. **f** Photovoltage–position curve obtained when different positions of the sample were irradiated by a laser with wavelength of 532 nm and power of 236 mW, in which the position 0 mm is the central position of the heterojunction.

where $k_B$ is the Boltzmann constant, $T$ is temperature (in this case, room temperature), and $R$ is the channel resistance of the device.

The specific detectivity ($D^*$), which is also called the detection sensitivity, is calculated by the following equation:

$$D^* = \frac{\sqrt{S_0}}{\text{NEP}} \quad (4)$$

where $S_0$ is the area of the light spot on the heterojunction.

From the curves in Fig. 3d-e, the AgNSF/CNTF heterojunction shows a significant photoelectric response from ultraviolet to terahertz wavelengths. The device has an output photovoltage of 11.63 mV when the AgNSF/CNTF heterojunction was irradiated by a laser with wavelength of 1064 nm and a power of 518 mW.

The calculated results of the key performance parameters for photoelectric conversion of the AgNSF/CNTF are given in supplementary Table 3. The photoelectric responsivity $R_V$ ranges from 9.35 to 40.4 mV W$^{-1}$, and the response time is in tens of milliseconds except for the case with irradiation by a laser with wavelength of 10.6 μm and 118.8 μm. The maximum photoelectric responsivity reaches 40.4 mV W$^{-1}$ for the case irradiated by a laser with wavelength of 405 nm and power of 163.2 mW. When the sample is irradiated by a laser with wavelength of 118.8 μm, the response time reaches an order of seconds, which might derive from the strong absorption of photons at low energy and poor thermal conductivity of the glass substrate. The equivalent noise power of the device was on the order of $10^{-8}$ W Hz$^{-0.5}$, and the detectivity was on the order of $10^7$ jones.

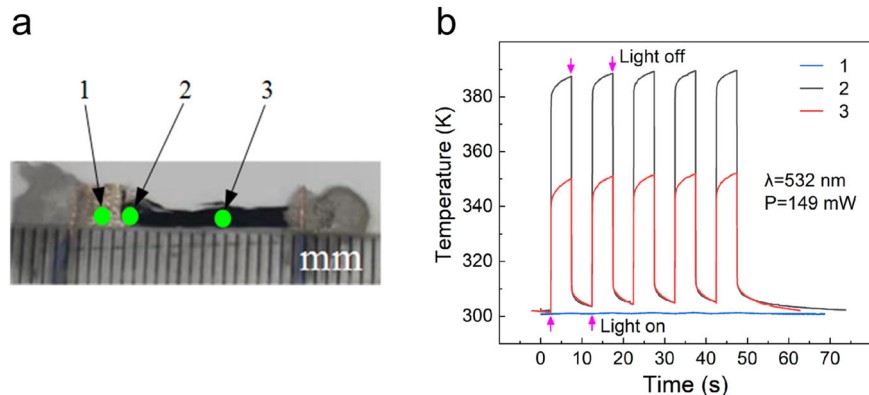

**Fig. 4 Comparison of the photothermal conversion ability of the AgNSF, AgNSF/CNTF heterojunction, and CNTF.** AgNSF (test point 1), AgNSF/CNTF heterojunction (test point 2), and CNTF (test point 3) of the sample irradiated by a laser with wavelength of 532 nm, power of 149.0 mW, and spot diameter of 1.2 mm. **a** Optical photograph of the actual sample. **b** Temperature–time response curves of the test points.

**Table 1 Comparison of the photothermal conversion performance of the AgNSF/CNTF heterojunction, AgNSF, and CNTF.**

| Wavelength (nm) | Incident light power (mW) | Rise temperature of heterojunction $\Delta T_1$ (K) | Rise temperature of AgNSF $\Delta T_2$ (K) | Rise temperature of CNTF $\Delta T_3$ (K) | $\frac{\Delta T_1 - \Delta T_3}{\Delta T_3}$ |
|---|---|---|---|---|---|
| 405 | 120.0 | 70.6 | 2.7 | 42.0 | 68.1% |
| 532 | 149.0 | 85.4 | 0.5 | 48.6 | 75.7% |
| 633 | 97.4 | 36.8 | 0.3 | 20.4 | 80.4% |
| 1064 | 163.0 | 69.9 | 0.1 | 44.4 | 57.4% |

A laser with wavelength of 532 nm and power of 236 mW is used to successively irradiate different positions of the sample. The curve of the change in photovoltage with the position is shown in Fig. 3f. The maximum photovoltage is generated when the laser irradiated the center of the heterojunction, while the photovoltage rapidly decreased when the laser irradiated both sides of the heterojunction. In addition, when the light spot irradiated the AgNSF side, the photovoltage decreased slightly faster with the position than when the light spot irradiated the CNTF side, and the photovoltage finally decreases to nearly 0 mV.

## Discussion

We compare the photothermal conversion ability of the AgNSF, AgNSF/CNTF heterojunction, and CNTF. A laser with incident light power of 149 mW, wavelength of 532 nm, and spot diameter of 1.2 mm is used to irradiate the AgNSF (region 1), AgNSF/CNTF heterojunction (region 2), and CNTF (region 3) in Fig. 4a. The temperature–time response curves are shown in Fig. 4b.

When the test points were irradiated by a 532 nm laser, the temperatures of the AgNSF, AgNSF/CNTF heterojunction, and CNTF increased by 0.5, 85.4, and 48.6 K, respectively (Fig. 4b). The similar test is also performed with 405, 633, and 1064 nm lasers. The temperature–time response curves are shown in Supplementary Fig. 1. Table 1 provides the corresponding comparison results. According to the data in Table 1, the photothermal conversion ability of the AgNSF/CNT heterostructure is significantly enhanced compared with that of the pure CNTF. In general, the rise temperature of the AgNSF/CNTF heterojunction was more than 50% higher than that of the pure CNTF, while the surface temperature of the AgNSF did not significantly increase when irradiated by the laser. Our explanation for this phenomenon is as follows. As shown in Supplementary Fig. 3a, the AgNSF is mainly composed of pure silver nanoparticles and nanowires with diameters of 40–100 nm. When a laser irradiates the AgNSF, it produces plasmons that propagate along the surface of the silver nanowires[17]. Ag is a good

heat conductor, so it will not produce an obvious local high-temperature region on the surface of the AgNSF. However, when the laser irradiates the AgNSF/CNTF heterojunction, it can produce localized surface plasmons in the interface of the AgNSF and CNTF. On the one hand, the local surface plasmons can effectively promote absorption of photons by the CNTF [31]. On the other hand, the coupling between the plasmons and the silver nanoparticles can form nanoscale "hot spots", which can increase the intensity of the laser local field irradiated on the heterojunction by more than four orders of magnitude in a very small area[32]. Therefore, the CNTF absorb more photons in the local micro/nano size region when the AgNSF is spread underneath, and the local temperature increases rapidly and significantly. In addition, other phenomena could also conduct to the temperature increase, like a reduction of the cross-plane thermal conductivity as already reported in layered structures[33].

From the Temperature–position curves in Fig. 1b, c, when a laser irradiates the heterojunction of the sample, the temperature of the heterojunction is significantly higher than those of the AgNSF and CNTF at the contact ends with the electrodes. The CNTF consists of metallic and semiconducting nanotubes[34], in which there are two types of charge carriers: electrons and holes. The temperature of the heterojunction after laser irradiation was significantly higher than the temperatures at the far ends of the AgNSF and CNTF (Fig. 1). For the CNTF (see Fig. 3a), the holes in the CNTs at the heterojunction have higher energy and spread to the area with lower temperature. As the holes spread to the cold end of the CNTF, more and more holes accumulate at the cold end. However, there are more negative charges at the heterojunction, and the built-in electric field formed prevents the holes from moving to the cold end. When the internal electric field force received by the holes and the driving force provided by the $\Delta_T$ reach a balance, the internal electric field formed in the CNTF points from the CNTF to the heterojunction. According to previous reports[35], electrons flow from the CNTs to the metal when a laser irradiates a heterojunction composed of CNTs and a

metal. Therefore, while the holes move from the heterojunction to the cold end of the CNTF, the electrons move from the CNTF to the AgNSF.

A more visual way to study the photothermoelectric effect is to compare the photovoltage–time and temperature–time response curves obtained at the same step in the same coordinate system. If they show the same trend and the response time is on the same order of magnitude, the photoelectric response behavior of the sample is mainly caused by the Seebeck effect. Bao's group used this method to show one period when Ag/SrTiO₃ heterostructure is irradiated by the laser with a wavelength of 10.57 μm[36]. To make a detailed comparison between the photovoltage–time and temperature–time response curves, we set the switching period of the optical shutter to 1, 2, 20, and 100 s and then irradiated the heterojunction with a laser with wavelength of 532 nm and power of 149 mW. The response curves of the photovoltage and temperature with time were synchronously recorded. All of the temperatures in the temperature–time curves were then subtracted from the initial temperature of the sample to obtain the $\Delta T$–time curves. The photovoltage–time and $\Delta T$–time curves plotted in the same coordinate system for visual comparison are shown in Fig. 5. Figure 5 intuitively and clearly indicates that the photovoltage–time and $\Delta T$–time curves show the same dynamic response law and basically the same response time and numerical variation trend, further confirming that the output photovoltage of the sample is mainly generated by the Seebeck effect.

The $\Delta U$–laser power and corresponding $\Delta_T$–laser power curves in the same coordinate system for laser wavelengths of 405, 532, 633, and 1064 nm are shown in Fig. 6, where $\Delta U$ is the increment of the photovoltage, $\Delta_T$ is the temperature difference between the heterojunction and far end of the CNTF, and the laser power refers to the absorption power of the sample.

From Fig. 6, when lasers with the same wavelength but different laser power irradiated the AgNSF/CNTF heterojunction,

the $\Delta U$ and $\Delta_T$ values at the heterojunction approximately linearly increased with increasing absorbed laser power, indicating that the device has stable photoelectric response and photothermal conversion performance with different laser power.

As shown in Fig. 7a (the data points here are derived from Figs. 2 and 6), we obtain Seebeck coefficient $S$ of the CNTF by fitting the data points, which is $S = +56.6 \, \mu V \, K^{-1}$ (according to a literature review, the Seebeck coefficient of the AgNSF is approximately $S_{AgNSF} = +1 \, \mu V \, K^{-1}$ [37,38], which can be ignored here). From Fig. 7a, except for the laser with wavelength of 375 nm (the data point indicated by a black arrow), when the heterojunction was irradiated by the lasers with the other wavelengths, the experimental data points are all on the straight fitting line, or distributed on both sides of the straight fitting line very closely. Our explanation is as follows, as shown in Fig. 7b, the absorption peak of CNTF appears at 270 nm, when the wavelength of incident light increases, the absorption rate decreases, as the wavelength of incident light approaches 270 nm, the interband transition of electrons corresponding to the photoelectric effect in CNTF enhances significantly. Therefore, when the heterojunction is irradiated by the laser with the wavelength of 375 nm, the proportion of photoelectric effect in the contributed photovoltage increases obviously. The above results indicated that the photovoltage generated by laser irradiation of the AgNSF/ CNTF heterojunction in the wavelength range from visible to terahertz is mainly contributed by the photothermoelectric effect.

In conclusion, the AgNSF/CNTF van der Waals heterojunction showed excellent photothermal conversion performance. When the heterojunction of the device was irradiated by different lasers with ultrawide-band range from ultraviolet to terahertz wavelengths, the temperature increased obviously and rapidly. The results of comprehensive evaluation based on multiple parameters in the Supplementary Table 1 clearly show that our device with AgNSF/CNTF heterojunction is superior to other devices in

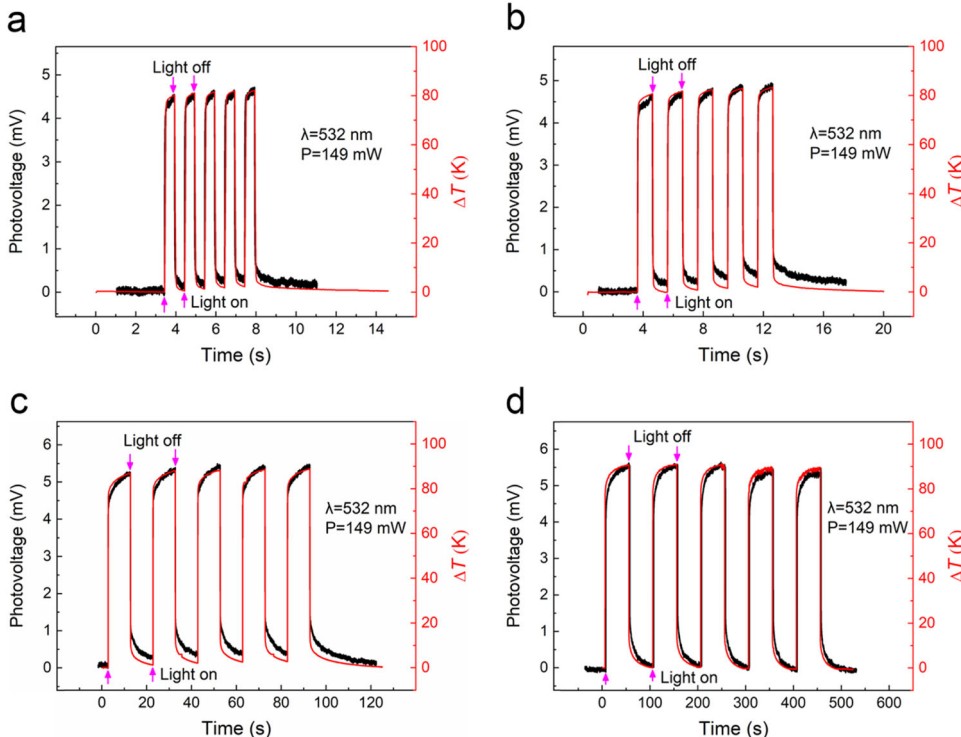

**Fig. 5 Comparison of photovoltage–time and $\Delta T$-time curves. a–d** Photovoltage–time and $\Delta T$–time curves on the same time axis when the heterojunction of the sample was irradiated by a laser with wavelength of 532 nm and power of 149 mW, and the cycle of the optical shutter switching was set to 1, 2, 20, and 100 s, respectively.

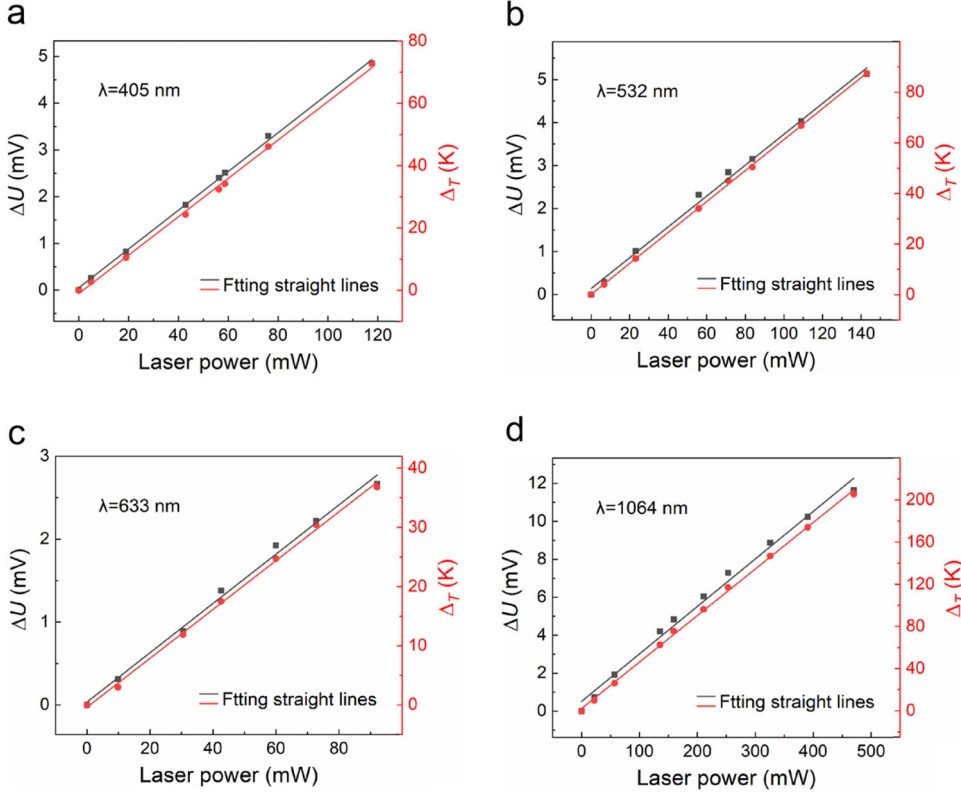

**Fig. 6 The relationship between $\Delta U$, $\Delta_T$ and Laser power. a–d** $\Delta U$–laser power and $\Delta_T$–laser power curves measured when the AgNSF/CNTF heterojunction was irradiated by lasers with wavelengths of **a** 405, **b** 532, **c** 633, and **d** 1064 nm, respectively.

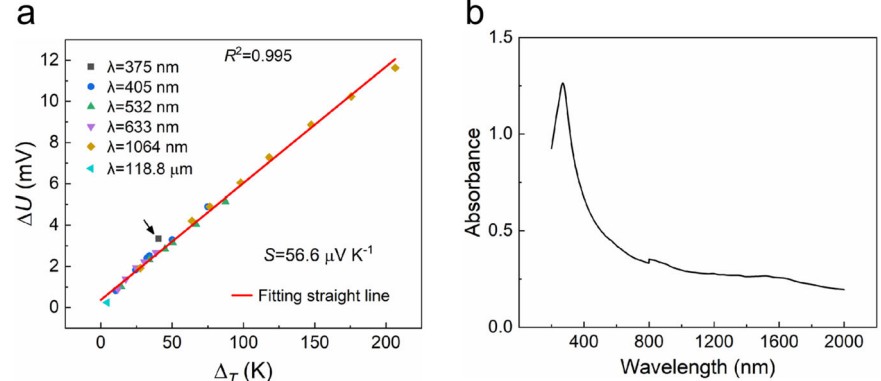

**Fig. 7 Thermal and optical properties of CNTF. a** The $\Delta U$ across the sample versus the corresponding $\Delta_T$ to determine the Seebeck coefficient ($S$) of CNTF. **b** Absorption spectrum of CNTF.

photothermoelectric effect. The photothermal responsivity was in the range 175–601 K W$^{-1}$, and the response time was on the order of tens of milliseconds (see Supplementary Table 2). Local surface plasmons can enhance light absorption of the CNT material. When the pure CNTF and AgNSF/CNTF heterostructure were irradiated with 405, 532, 633, and 1064 nm lasers, the variation of the increase of the temperature of the AgNSF/CNTF increased by 57.4–80.4% compared with the CNTF (see Table 1). The AgNSF/CNTF heterojunction also showed a significant photoelectric response in the ultrawide band from ultraviolet to terahertz wavelengths. Except for terahertz and mid-infrared wavelengths, the response time was on an order of tens of milliseconds, and the light responsivity was tens of millivolts per watt. Seebeck effect plays a leading role in photovoltage generation, including some photoelectric effects related to incident laser wavelength. The AgNSF/CNTF heterojunction can not

only be made into a fast-response ultra-wideband photodetector, but also can be used as a high-efficiency sensitive thermal collector. The AgNSF/CNTF heterostructure has good application prospects as a high-quality photothermoelectric material.

## Methods

**Preparation of the sample**. We first prepared a AgNSF by a solid-state ionics method[13]. A mask with a length of 2 cm was placed on the glass substrate. Under the conditions of temperature of 30 °C and vacuum degree of $3 \times 10^{-3}$ Pa, 880 mg of silver was evaporated on the glass substrate by vacuum thermal evaporation to form two Ag film electrodes with a spacing of 2 cm. A layer of RbAg$_4$I$_5$ film with a thickness of hundreds of nanometers was evaporated on the entire substrate under the same conditions. Using a Keithley 2400 source meter, a constant current of 1 μA was applied to the two silver-film electrodes and the samples were continuously grown for 98 h, and the AgNSF was obtained. The CNTF was prepared by chemical vapor deposition (CVD)[34]. The parameters were an argon gas flow rate of 2000 mL min$^{-1}$, a hydrogen flow rate of 600 mL min$^{-1}$, a carbon-source flow

rate of 28 μL min$^{-1}$, normal pressure, and reaction temperature of 1180 °C. The prepared CNTF was a mixture of single- and double-walled CNTs[34].

The process for preparing the AgNSF/CNTF van der Waals heterojunction is shown in Supplementary Fig. 2. First, a clean glass was chosen as substrate. A small piece of AgNSF was then cut, transferred to the glass substrate with tweezers. A drop of anhydrous ethanol was added to attach the AgNSF to the surface of the glass substrate (Supplementary Fig. 2a). A small piece of the CNTF was then cut, transferred to the glass substrate with tweezers, and its left end was overlapped with the upper surface of the right end of the AgNSF. A drop of anhydrous ethanol was then added. When the anhydrous ethanol evaporated, the van der Waals force bound the CNTF and AgNSF together, and the AgNSF/CNTF van der Waals heterojunction formed (Supplementary Fig. 2b). Finally, wires were drawn from the other ends of the AgNSF and CNTF, and then the wires were fixed with silver paint (Supplementary Fig. 2c). The overlapped area between the AgNSF and the CNTF was about 1.5 × 2.0 mm. A low-magnification scanning electron microscopy image of the heterojunction is shown in Supplementary Fig. 2d. An atomic force microscopy image of the prepared sample showed that the average thickness of the AgNSF was about 220 nm, the average thickness of the CNTF was about 200 nm.

**Morphological characterization and elemental analysis of the sample**. We performed field emission scanning electron microscopy with the energy-dispersive spectroscopy (EDS) analysis function (Zeiss Merlin) to characterize the AgNSF, CNTF, and AgNSF/CNTF heterojunction part of the two overlapped structures (Supplementary Fig. 3a). In the high-magnification scanning electron microscopy image, the grown AgNSF was a tightly arranged Ag nanowire cluster structure (region I in Supplementary Fig. 3a). The white particles indicated by the black arrows are silver nanoparticles, and the diameter of the silver nanowires was about 40–100 nm. The EDS spectrum showed that the composition of the AgNSF in the I region was pure silver (Supplementary Fig. 3b). The heterojunction between the AgNSF and CNTF is shown in the region II in Supplementary Fig. 3a. The EDS spectrum of region II showed that carbon, iron, and silver were present in the heterojunction (Supplementary Fig. 3c). The CNTF was a network structure composed of a series of CNT bundles with different directions (region III in Supplementary Fig. 3a). The diameters of the CNT bundles were about 6–9 nm. The white arrows in Supplementary Fig. 3a indicate the catalyst iron particles left after growth of the CNTs by the CVD method[34]. The EDS spectrum of region III showed that the CNTF mainly contained carbon and a small amount of iron (Supplementary Fig. 3d).

**Voltage–current characteristics measurement**. Figure 3a is the setup used to measure the current–voltage curves of the sample, the AgNSF was connected to the positive electrode of a Keithley 2400 source meter and the CNTF was connected to negative electrode of the source meter. At room temperature and atmospheric conditions, the heterojunction was irradiated with lasers with different wavelengths (wavelength 375 nm, spot diameter 2.5 mm, output power 110 mW; wavelength 405 nm, spot diameter 1.5 mm, output power 169 mW; wavelength 532 nm, spot diameter 1.2 mm, output power 246.8 mW; wavelength 633 nm, spot diameter 2.5 mm, output power 97.4 mW; wavelength 1064 nm, spot diameter 2.0 mm, output power 523 mW; wavelength 10.6 μm, spot diameter 2.0 mm, output power 160 mW; and wavelength 118.8 μm, spot diameter 1.5 mm, output power 75 mW) and obtained the current–voltage curves. When measuring the current–voltage curves, a scanning voltage range was from −10 to +10 mV.

**Photothermal and photoelectrical response performance test**. To explore the photoelectric response performance of the device, we irradiated the AgNSF/CNTF sample with lasers with wavelengths of 375 nm, 405 nm, 532 nm, 633 nm, 1064 nm, 10.6 μm, and 118.8 μm. The continuous laser was modulated into periodic inter-mittent light of a certain frequency through an optical shutter (Supplementary Fig. 4). The photovoltage generated at both ends of the sample with and without illumination were then recorded by a Tektronix DPO 5104B high-frequency oscilloscope, and the photovoltage–time response curves were plotted. To make the photovoltage–time curves more accurate and completely reflect the photoelectric response process of the sample, we set the cycle of optical shutter switching for laser irradiation to 10 s (for terahertz wave irradiation, it takes a long time to achieve a stable photovoltage output, so we set the cycle of optical shutter switching to 2 min). At the same time, an Optris PI 640 infrared camera was used to syn-chronize the video of the thermal distribution on the surface of the sample with and without light to record the change of the surface temperature of the sample with time (see Supplementary Information 1). The end of the CNTF of the sample was connected to the positive electrode of the oscilloscope port, and the end of the AgNSF was connected to the negative electrode of the oscilloscope port.

**The calibration of temperature measurement**. We calibrate the temperature measurement of the infrared camera by using a thermalcouple and separately heating the AgNSF, CNTF, and AgNSF/CNTF heterojonction samples with a hot plate. The emissivity values of the samples are thus adjusted to make the tem-perature measured with the infrared camera equal to that with the thermocouple. The emissivity is 0.77, 0.98, and 0.97 for the AgNSF, CNTF, and the AgNSF/CNTF heterojunction samples, respectively.

## Data availability

Source data are provided with this paper.

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

## Acknowledgements

This work was partially supported by the NSAF (Grant No. U1730246), the Open Research Fund Program of the State Key Laboratory of Low-Dimensional Quantum Physics (No. KF202007), the Beijing Municipal Natural Science Foundation (Grant No. 4192024), and National Natural Science Foundation of China (Grant No. 51972188).

## Author contributions

B.L. and J.-L.S. designed and prepared the test sample. Y.L. designed a comparative experiment to explore the key roles of localized surface plasmon on the large temperature rise. B.L., W.W., Y.X., J.-L.Z., Y.C., W.M., J.W., and J.-L.S. performed the experiments and data analysis. B.L. and J.W. synthesized the AgNSF and CNTF, respectively. Y.J. was involved in calibration of the thermal camera and temperature measurement, as well as carrying on the comparative experiments. J.W. and J.-L.S. designed the project and guided the research. B.L. wrote the manuscript. J.-L.Z., N.Y., W.C., J.W., and J.-L.S. reviewed the manuscript.

## Competing interests

The authors declare no competing interests.
