## [Peer Review File · Nature Communications]

Local large temperature difference and ultra-wideband photothermoelectric response of the silver nanostructure film/carbon nanotube film heterostructureREVIEWER COMMENTS

Reviewer #1 (Remarks to the Author):

The AgNSF/CNTF van der Waals heterojunction showed excellent photothermal conversion performance. The local temperature difference and the output photovoltage increase rapidly when the heterojunction is irradiated by lasers with wavelengths ranging from ultraviolet to terahertz. The maximum of the temperature difference reaches 205.9 K, which is significantly higher than that of other photothermoelectric materials reported in literatures. The methods, including preparation of the sample, morphological characterization and elemental analysis of the sample, voltage–current characteristics measurement, and photothermal and photoelectrical response performance test, seem sound with many supplementary materials presented.

While there is a minor error, the maximum of the temperature difference reaches 215.9 K ($512.15\text{K}-296.25\text{K}=215.9\text{K}$), not 205.9 K. After this minor revision, it can be published.

Reviewer #2 (Remarks to the Author):

In this article, a heterojunction made of a joined silver nanostructure film (AgNSF) and a carbon nanotube film (CNTF) is investigated for its photothermoelectric properties. When irradiated by lasers with wavelength ranging from terahertz to ultraviolet, the local temperature difference reaches a maximum of 205.9 K, which is according to the authors the best value to date reported in the literature. A thermovoltage is then generated thanks to this temperature difference by Seebeck effect. Notably, a very fast response time is also reported. Seebeck effect as the main source of voltage in this system was proved by comparing the photovoltage-time and temperature-time responses.

Authors attribute these outstanding performances to the strong surface plasmon at the interface between AgNSF and CNTF heterojunction as attributed earlier by several research teams in other systems. As a consequence, CNTF monolayer absorbs more light when stacked on the AgNSF film compared with the CNT film alone. The potential applications are improved photodetectors and solar cell collectors. The article is well written, interesting and open fruitful perspectives in energy conversion. Sample preparation and experiments are correctly described. However, some points deserve clarification:

1- The main contribution of this article is the high local temperature difference (205.9 K) induced by illumination, more important than those reported in other studies. Considering this metric as indicated in supplementary table 1, results reported for AgNSF/CNTF heterojunction are the best to date. Nevertheless, this temperature elevation depends on the laser light power density. Is the local temperature difference elevation temperature per unit of power density would not be a better performance indicator as compared to the temperature alone for assessing different photothermoelectric converters?

2- Temperature elevation was determined by an infrared camera. Measured temperature accuracy depends critically on the material emissivity added as an input parameter in the camera. Is the CNT film emissivity was taken into account in the calibration? How was determined the CNT film emissivity? Due to plasmonic effect are emissivities of a single CNT film and AgNSF/CNTF heterojunction similar?

3- The large temperature increase is attributed to a plasmonic effect as mentioned by several authors for others nanolayered structures. No experimental evidence supporting this assumption is given in this study. Other phenomena could conduct to same behaviour like a reduction of the cross-plane thermal conductivity as already reported in layered structures [Chiritescu et al.]. Perhaps investigating the thermal conductivity in AgNSF/CNTF heterojunctions could give additional insight and allow to discriminate between the contribution of the plasmonic effect with a possible thermal conductivity reduction.

Chiritescu et al. Ultralow Thermal Conductivity in Disordered, Layered WSe₂ Crystals, *Science* 315, 351 (2007);

December 27, 2021

Dear Reviewers,

Thank you for your comments concerning our manuscript entitled “*Local large temperature difference and ultra-wideband photothermoelectric response of the silver nanostructure film/carbon nanotube film heterostructure*” (Manuscript ID: **NCOMMS-21-38942B**).

The comments are valuable and helpful for improving our manuscript. We have made careful revisions of the manuscript according to your comments and the editorial requests. The main corrections in the revised manuscript and the point-by-point responses to your comments are included in the following. We highlighted the changes in yellow in the revised manuscript. Thank you very much for your consideration.

Sincerely yours,

Prof. Jia-Lin Sun

Address: Department of Physics, Tsinghua University, Beijing 100084, P.
R. China.

ORCID ID: 0000-0002-1849-5916

Responses to the reviewers' comments:

Reviewer 1:

Comment: The AgNSF/CNTF van der Waals heterojunction showed excellent photothermal conversion performance. The local temperature difference and the output photovoltage increase rapidly when the heterojunction is irradiated by lasers with wavelengths ranging from ultraviolet to terahertz. The maximum of the temperature difference reaches 205.9 K, which is significantly higher than that of other photothermoelectric materials reported in literatures. The methods, including preparation of the sample, morphological characterization and elemental analysis of the sample, voltage–current characteristics measurement, and photothermal and photoelectrical response performance test, seem sound with many supplementary materials presented.

While there is a minor error, the maximum of the temperature difference reaches 215.9 K ($512.15\text{K}-296.25\text{K}=215.9\text{K}$), not 205.9 K. After this minor revision, it can be published.

Response:

Thank you very much for your positive comments on our work. We considered the experimental results carefully, and believed that the maximum of the temperature difference reaches 215.9 K. It is our carelessness to make such a mistake. We have made corresponding correction in our revised manuscript.

Reviewer 2:

Comment: In this article, a heterojunction made of a joined silver nanostructure film (AgNSF) and a carbon nanotube film (CNTF) is investigated for its photothermoelectric properties. When irradiated by lasers with wavelength ranging from terahertz to ultraviolet, the local temperature difference reaches a maximum of 205.9 K, which is according to the authors the best value to date reported in the literature. A thermovoltage is then generated thanks to this temperature difference by Seebeck effect. Notably, a very fast response time is also reported. Seebeck effect as the main source of voltage in this system was proved by comparing the photovoltage-time and temperature-time responses.

Authors attribute these outstanding performances to the strong surface plasmon at the interface between AgNSF and CNTF heterojunction as attributed earlier by several research teams in other systems. As a consequence, CNTF monolayer absorbs more light when stacked on the AgNSF film compared with the CNT film alone. The potential applications are improved photodetectors and solar cell collectors. The article is well written, interesting and open fruitful perspectives in energy conversion. Sample preparation and experiments are correctly described. However, some points deserve clarification:

Response:

We appreciate your comments which help us to improve the quality and scientific significance of our manuscript. We have revised the manuscript according to your comments and suggestions.

Comment: 1- The main contribution of this article is the high local temperature difference (205.9 K) induced by illumination, more important than those reported in other studies. Considering this metric as indicated in supplementary table 1, results reported for AgNSF/CNTF heterojunction are the best to date. Nevertheless, this temperature elevation depends on the laser light power density. Is the local temperature difference elevation temperature per unit of power density would not be a better performance indicator as compared to the temperature alone for assessing different photothermoelectric converters?

Response:

We agreed to your opinion that the temperature elevation depends on the power density of laser. We found that the temperature elevation also depends on the time of laser illumination. Therefore, we added two columns of "The change in temperature per unit power density " and "The change in temperature per unit power density and per unit time" in the **Supplementary Table 1** to assess different photothermoelectric converters. The results clearly show that our device with AgNSF/CNTF heterojunction is superior to other devices in photothermoelectric effect. We added some description of the photothermoelectric effect in "**Conclusion**" in our revised manuscript.

Comment: 2- Temperature elevation was determined by an infrared camera. Measured temperature accuracy depends critically on the material emissivity added as an input parameter in the camera. Is the CNT film emissivity was taken into account in the calibration? How was determined the CNT film emissivity? Due to plasmonic effect are emissivities of a single CNT film and AgNSF/CNTF heterojunction similar?

Response:

Thank you for your comments. We believe that it is the best method to measure the localized temperature of the samples illuminated by a laser in vacuum chamber through infrared camera. In our previous temperature measurement, we have calibrated the parameters of the infrared camera and used an emissivity of 0.98 for the CNTF, and 0.97 for the AgNSF/CNTF heterojunction.

Here, we re-calibrated the temperature measurement of the infrared camera by using a thermocouple and separately heating the CNTF and AgNSF/CNTF heterojunction samples with a hot plate. The emissivity values of the samples are thus adjusted to make the temperature values measured with the infrared camera equal to those measured with the thermocouple. The recalibrated emissivity is 0.98 for the CNTF sample, and is 0.97 for the AgNSF/CNTF heterojunction, which are exactly the same values we input in the infrared camera in our previous experiments. We added a calibration method of "**The calibration of temperature measurement.....**" in the "**Method**" section in our revised manuscript.

Comment: 3- The large temperature increase is attributed to a plasmonic effect as mentioned by several authors for others nanolayered structures. No experimental evidence supporting this assumption is given in this study. Other phenomena could conduct to same behaviour like a reduction of the cross-plane thermal conductivity as already reported in layered structures [Chiritescu et al.]. Perhaps investigating the thermal conductivity in AgNSF/CNTF heterojunctions could give additional insight and allow to discriminate between the contribution of the plasmonic effect with a possible thermal conductivity reduction.

Chiritescu et al. Ultralow Thermal Conductivity in Disordered, Layered WSe₂ Crystals, Science 315, 351 (2007);

Response:

We agreed to your opinion on that other authors have reported large temperature rise induced by the plasmonic effects.

We believe that the large temperature rise here is mainly attributed to the plasmonic effect. We also believe that other phenomena might also conduct to similar behaviors, such as a reduction of the cross-plane and heterojunction thermal conductivity. It might be able to understand the key factors on the large temperature rise by measuring the thermal conductivity in the AgNSF/CNTF heterojunction with and without light illumination, or examining the two samples with strong and weak plasmonic effects. However, it is not easy to measure the thermal conductivity of the AgNSF/CNTF heterojunction, which is affected by many factors, such as CNTF, AgNSF, interface between the two components, and substrate. Furthermore, the thermal conductivity of the heterojunction might change due to plasmonic effect under light illumination.

Alternatively, we designed a comparative experiment by using two devices with similar structure: AgNSF/CNTF and Ag film/CNTF. The two devices have almost equal thermal conductivity, but have different plasmonic effects under light illumination. As shown in **Figure R1**, we first deposited a thin Ag film (~100 nm in thickness) on one side of a glass slide through vacuum thermal evaporation. Then, we transferred a AgNSF (~220 nm in thickness) to the other side of the glass slide. After adding a few drops of ethanol, the AgNSF stick tightly on the glass substrate. The plasmonic effect of the Ag film and AgNSF were characterized through surface-enhanced Raman scattering (SERS) spectra using rhodamine 6G (R6G) as probe molecule. After performing the SERS spectra, we transferred a large CNT film which can cover both of the Ag film and AgNSF, and made the CNTF contact with the Ag film and AgNSF through van der Waals interaction by dropping several ethanol drops.

The surface morphology of the Ag film and AgNSF was characterized through a multifunctional and high-resolution optical and spectroscopic imaging microscope (Alpha 300RAS, WITec). **Figure R2a** and **b** are microscopic images of the AgNSF and Ag film, respectively. The AgNSF consists many aligned nanowire clusters, while the overall surface of Ag film is relatively flat. Both of the silver and CNT materials have high thermal conductivity. Since the AgNSF is one-fold thicker than the Ag film, we believe that the thermal conductivity in dark of the AgNSF/CNTF device is equivalent to or even higher than that of the Ag film/CNTF one. Thus, our experiment can eliminate the effects of homogeneity of CNTF and thermal conductivity on the temperature rise.

Figure R2c and **d** are SERS spectra of the R6G probe at different concentrations on the AgNSF and Ag film substrates. It is clear that Raman spectra of the R6G are enhanced significant for both of the AgNSF and Ag film. The AgNSF has a detect limit of the R6G at least of 10^{-8} M. But the Ag film can detect R6G solution with concentration at only about 10^{-5} M. It implies that the AgNSF has stronger SERS effect than that of the Ag film. According to literature, the SERS is mainly contributed to the plasmonic effects of the substrate [**Le Ru, E.C. et al. Surface enhanced Raman spectroscopy on nanolithography-prepared substrates. *Curr. Appl. Phys.* **8**, 467 (2008)**]. It implies that the AgNSF has much stronger plasmonic effect than that of the Ag film when illuminated by laser.

Here, we chose two kinds of lasers (532 nm, 224 mW and 1064 nm, 518 mW) to irradiate these two devices (the green circle spot represents the light spot in **Figure R1d**), and tested the local temperature differences generated after the same irradiation time. The results show that the local temperature differences are 90.4 °C (**Figure R3a**) and 138.8 °C (**Figure R3b**) when the CNTF/Ag film/glass was irradiated with 532 nm and 1064 nm lasers, respectively. Irradiation on CNTF/AgNSF/glass with 532 nm and 1064 nm lasers resulted in temperature rises of 127.9 °C (**Figure R3c**) and 209.0 °C (**Figure R3d**), respectively. It can be seen that under a same laser irradiation, the local temperature difference of CNTF/AgNSF is significantly higher than that of the CNTF/Ag film. By combining the SERS results of the Ag film and AgNSF samples, we believe that the plasmonic effect of the AgNSF play a key role on the large temperature rise. We believe that the local surface plasmons can significantly promote the conversion of light energy to heat energy, which have been reported in literatures [**Bell, A.P. et al. Quantitative study of the photothermal properties of metallic nanowire networks. *ACS Nano* 9, 5551-5558 (2015); Brongersma, M.L. et al. Plasmon-induced hot carrier science and technology. *Nat. Nanotech.* 10, 25-34 (2015)**].

Therefore, we made some modification by adding a sentence "In addition, other phenomena could also conduct to the temperature increase like a reduction of the cross-plane thermal conductivity as already reported in layered structures³³" in Page 8 in the revised manuscript. We also added a reference mentioned above (Chiritescu et al. Ultralow Thermal Conductivity in Disordered, Layered WSe₂ Crystals, *Science* 315, 351 (2007)) in our revised edition.

Figure R1. Fabrication and performance test flow chart of comparative samples. **(a)** Ag film was thermally deposited on the right side of the glass substrate. **(b)** AgNSF was transferred to the left side of the same glass, and a few drops of ethanol was added to make it stick tightly on the glass surface. **(c)** A droplet of R6G aqueous solution with a concentration of $10^{-8} \text{ mol L}^{-1}$ were subsequently dropped on the surface of AgNSF and Ag film respectively in local area. After detecting the SERS spectra of the two samples, droplets of R6G aqueous solution with concentration of $10^{-5} \text{ mol L}^{-1}$ were dropped on the surface of AgNSF and Ag film in local area respectively, and the SERS spectra of the two samples were detected again. **(d)** Finally, a large CNTF was transferred to the surface of AgNSF and Ag film at the same time. Ethanol was dropped to make CNTF cover the surface of AgNSF and Ag film closely, forming two similar heterostructures in the central region of the glass substrate.

Figure R2. Micrographs of (a) AgNSF and (b) Ag film on glass substrate. (c) and (d) are SERS spectra of R6G from different samples and concentrations. First, R6G aqueous solution with concentration of 10^{-8} mol L $^{-1}$ was dropped onto the two samples. Two small regions marked with blue and black boxes were selected to test the SERS spectra of multiple points, and the average SERS spectra were obtained as shown in blue and black curves in subgraph c. Then, R6G aqueous solution with concentration of 10^{-5} mol L $^{-1}$ was dropped on the local area of the surface of the two samples again. Two small regions within the red boxes were selected to test the SERS spectra of multiple points, and the average SERS spectra were obtained as shown in the red curves in subgraphs c and d. SERS spectra were measured using 532-nm laser with a 50 \times objective under a power of 1 mW, the number of accumulations was 20, and the integration time was 1 s.

Figure R3. Local temperature differences and thermal distribution of the CNTF/Ag film and CNTF/AgNSF heterostructures irradiated by lasers with two wavelengths of 532 nm (a and c) and 1064 nm (b and d). a and b are the CNTF/Ag film heterostructure sample. c and d are the CNTF/AgNSF heterostructure sample.

REVIEWERS' COMMENTS

Reviewer #1 (Remarks to the Author):

I,m satisfied with the revisions. It can be published.

Reviewer #2 (Remarks to the Author):

Satisfactory and convincing responses to my queries were given by the authors. I notably appreciate the comparisons done between different works with results normalized in terms of "change in temperature per unit power density" and "the change in temperature per unit power density and per unit of time". That's why I recommand this article entitled "Local large temperature difference and ultra-wideband photothermoelectric response of the silver nanostructure film/carbon nanotube film heterostructure" for publication in Nature Communications.